# Synchronous ensembles of hippocampal CA1 pyramidal neurons during novel exploration

En-Li Chen[1,2], Tsai-Wen Chen[1,2], Eric R Schreiter[3], Bei-Jung Lin[1,2]*

[1]Institute of Neuroscience, National Yang Ming Chiao Tung University, Hsinchu, Taiwan; [2]Brain Research Center, National Yang Ming Chiao Tung University, Hsinchu, Taiwan; [3]Janelia Research Campus, Howard Hughes Medical Institute, Chevy Chase, United States

*For correspondence:
beijunglin@nycu.edu.tw

## eLife Assessment

In this **valuable** study, the authors use a cutting-edge method to perform voltage imaging of CA1 pyramidal cells in head-fixed mice running on a track while local field potentials (LFPs) were recorded in the contralateral hemisphere. The authors provide **solid** evidence of synchronous ensembles of CA1 pyramidal neurons that are associated with contralaterally recorded theta rhythms but not with contralaterally recorded sharp wave-ripples during exploration of a novel environment. The paper will be of interest to scientists who are interested in hippocampal neuronal coding of novel environments, particularly those with experimental questions that can benefit from this cutting-edge imaging technique.

**Abstract** Synchronous neuronal ensembles play a pivotal role in the consolidation of long-term memory in the hippocampus. However, their organization during the acquisition of spatial memory remains less clear. In this study, we used neuronal population voltage imaging to investigate the synchronization patterns of mice CA1 pyramidal neuronal ensembles during the exploration of a new environment, a critical phase for spatial memory acquisition. We found synchronous ensembles comprising approximately 40% of CA1 pyramidal neurons, firing simultaneously in brief windows (~25ms) during immobility and locomotion in novel exploration. Notably, these synchronous ensembles were not associated with contralateral ripple oscillations but were instead phase-locked to theta waves recorded in the contralateral CA1 region. Moreover, the subthreshold membrane potentials of neurons exhibited coherent intracellular theta oscillations with a depolarizing peak at the moment of synchrony. Among newly formed place cells, pairs with more robust synchronization during locomotion displayed more distinct place-specific activities. These findings underscore the role of synchronous ensembles in coordinating place cells of different place fields.

## Introduction

The establishment of long-term memory involves the initial acquisition and subsequent consolidation of memory traces, processes intricately connected to the modification of synaptic transmission (**Takeuchi et al., 2014**; **Moser et al., 2015**). Synchronous ensembles, characterized by the simultaneous firing of multiple neurons over tens of milliseconds (referred to as population synchrony), play a pivotal role in influencing synaptic plasticity (**Axmacher et al., 2006**; **Paulsen and Sejnowski, 2000**; **Wang, 2010**). The hippocampus, a crucial region for long-term memory formation (**Harris et al., 2003**; **Squire and Wixted, 2011**), has been a focal point of synchronous ensemble studies. Within

the hippocampus, synchronous ensembles are hypothesized to play roles in reactivating and consolidating labile memory traces (*Buzsáki, 2015*). However, their organization during memory acquisition remains less understood.

In the hippocampus, synchronous ensembles among CA1 pyramidal cells (CA1PCs) are predominantly observed during sharp-wave ripples, the brain waves associated with memory consolidation (*Buzsáki, 2015*; *Nádasdy et al., 1999*; *Cheng and Frank, 2008*; *Dragoi and Tonegawa, 2011*; *Liu et al., 2023*; *Colgin, 2016*). During these events, 10–18% of CA1PCs generate spikes, particularly during offline consummatory behavioral states or sleep (*Buzsáki and da Silva, 2012*; *Csicsvari et al., 1998*). These synchronous ensembles are biased toward experience-dependent reactivation, consolidating labile memory traces (*Gillespie et al., 2021*; *Yagi et al., 2023*). Furthermore, synchronous ensembles firing action potentials in short windows of 10–30 ms enhance information processing (*Harris et al., 2003*) and discriminate distinct behavioral contingencies (*El-Gaby et al., 2021*). Although synchronous ensembles of CA1PCs during offline memory consolidation are well-characterized, their organization during online acquisition of spatial memory, such as when animals enter a new environment and new place cells form, remains unclear.

Previous studies have identified correlated activities among place cells during theta oscillation, where 1–3% of place cells' spikes co-activate in a forward order matching animals' trajectory in the theta cycles when animals' locations overlap in the place fields of these cells (*Skaggs, 1996*; *Dragoi and Buzsáki, 2006*; *Foster and Wilson, 2007*). Because these place cells fire at distinct theta phases in the same cycle, this coordinated firing operates on the timescale of theta periods (~125ms; *Mizuseki and Buzsaki, 2014*; *O'Keefe and Recce, 1993*). Population synchrony on shorter timescales (10–30ms) is speculated to be reduced by the presence of theta oscillations (*Mizuseki and Buzsaki, 2014*).

The subthreshold membrane voltage (subVm), influenced by synaptic inputs and intrinsic conductances, directly triggers spiking activity (*Lee and Brecht, 2018*; *Petersen, 2017*). Additionally, properties of subVm support the organization of neuronal representation (*Giocomo et al., 2007*). Although correlated membrane potentials have been proposed to underlie the synchrony of neuronal ensembles (*Lampl et al., 1999*; *Poulet and Petersen, 2008*), multicellular recordings of subVms have not yet provided direct experimental evidence. Given the intricate interaction among subVm, spiking, and place cell formation, it is crucial to investigate the temporal relationships of sub- and suprathreshold neuronal activities among CA1PCs during novel exploration.

In this study, we used voltage imaging to investigate the temporal dynamics of sub- and suprathreshold neuronal activities in CA1PCs during novel exploration. Additionally, we implanted electrodes in the contralateral CA1 region to monitor theta and ripple oscillations. We found synchronous ensembles involving approximately 40% of CA1PCs exhibited transient population synchrony and correlated subthreshold membrane potentials when mice ran and stopped in a new maze. Moreover, these synchronous ensembles occurred outside of contralateral ripples (c-ripples) and were phase-locked to intracellular theta oscillations as well as extracellular theta oscillations recorded from the contralateral CA1 region. Notably, cell pairs with stronger synchronous activities established more distinct place fields that overlapped less. In essence, synchronous ensembles facilitated temporal association among place cells representing diverse features of spatial memory. This plastic network, driven by synchronous ensembles, may contribute to the stabilization of place cells, establishing place fields that effectively tile the environment.

## Results

To investigate the temporal relationships between neuronal activities during novel exploration, we simultaneously recorded many neurons as animals explored a new environment for the first time. To achieve this, we employed voltage imaging in conjunction with an air-lifted plastic track that allowed mice to explore an environment while remaining head-fixed under a microscope (*Figure 1A*). To ensure a genuinely novel experience, all pre-training and habituation were performed in a separate room distinct from the one where the imaging experiments were conducted. On the day of imaging, the mice were introduced to the recording room for the first time and initially confined to a specific corner of the track (*Figure 1A* top view). Subsequently, as the imaging session commenced, the confinement was lifted, granting mice the freedom to explore the track while imaging was conducted.

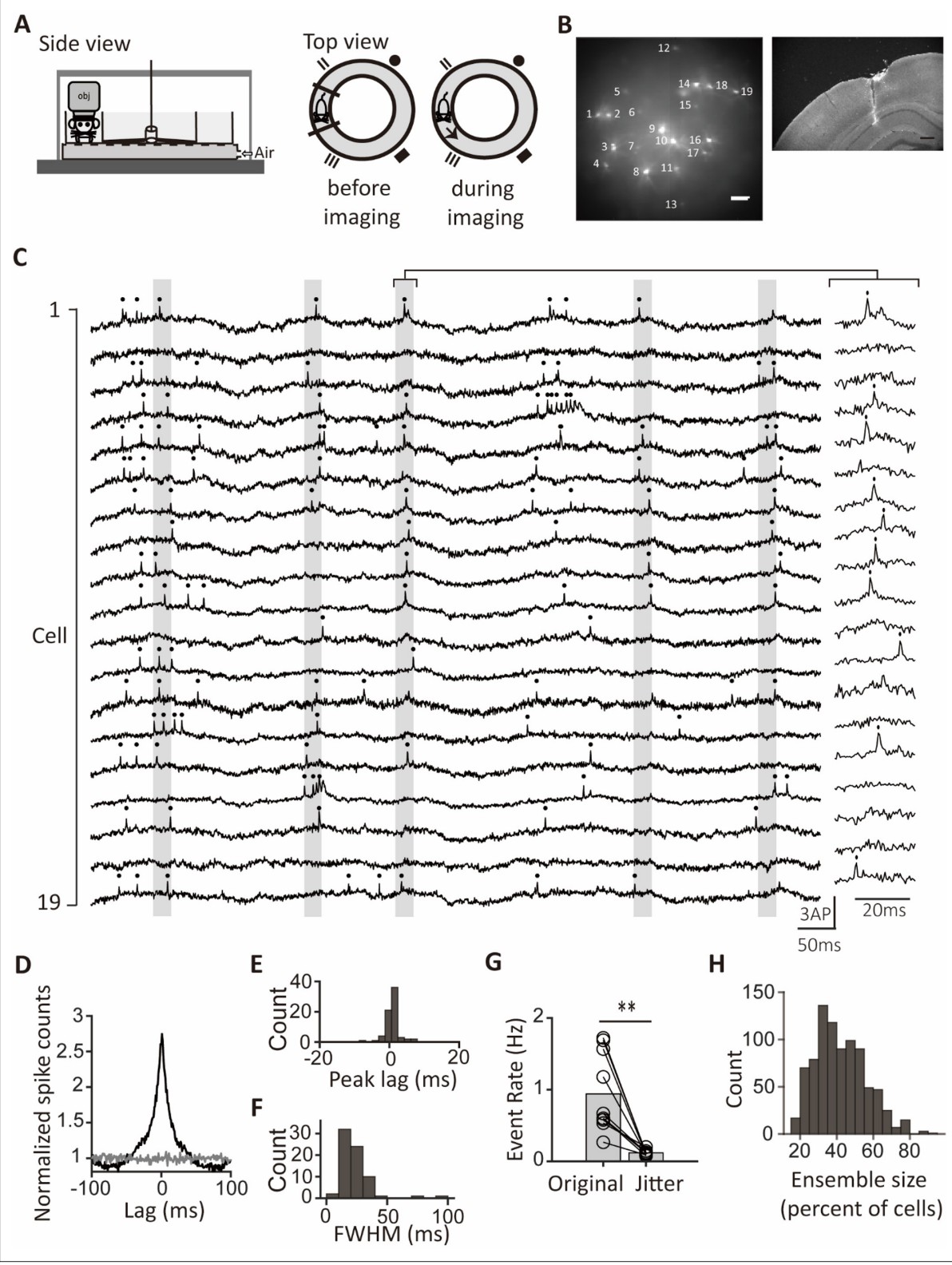

**Figure 1.** Population synchrony during novel exploration. (**A**) Schematic of the behavioral setup for voltage imaging. Mice explored a novel one-way track while fluorescence images were captured at the CA1 pyramidal layer. Left: side view; right: top view of the setup. (**B**) Left: example time-averaged image of CA1 pyramidal cells expressing Voltron2-ST labeled with JF$_{552}$-HaloTag ligand. Scale bar, 100 μm. Right: a microphotograph of a brain slice with the contralateral LFP electrode track from the same mouse as the left. Scale bar, 300 μm. (**C**) Fluorescence traces of cells in an example session with identified synchronous ensembles (labeled with gray vertical bars). Right: an example of magnified fluorescence traces within the bracket, pinpointing

*Figure 1 continued on next page*

*Figure 1 continued*

the occurrence of the identified synchronous ensemble. Spikes are indicated by dots on the top. (**D**) The grand average cross-correlogram (CCG) averaged across all 71 cells. The gray line represents the mean grand average CCG between reference cells and randomly selected cells from different sessions.(**E**) Distribution of the grand average CCG peak lags of all cells (n=71). (**F**) Distribution of the full width at half maximum (FWHM) of the grand average CCGs (n=71 cells). (**G**) Pairwise comparison of the event rates of population synchrony between original and jittered data. Bar heights indicate group means. **p<0.01 (**H**) Histogram of the ensemble sizes as percentages of cells participating in the synchronous ensembles (n=846 synchronous ensembles).

The online version of this article includes the following source data and figure supplement(s) for figure 1:

**Source data 1.** Excel file containing numerical data used to generate *Figure 1D-H*.

**Figure supplement 1.** Morphology of Voltron-expressing neurons and their positions along the deep-superficial axis.

**Figure supplement 2.** Paired photographs of the LFP electrode track and the voltage image from the same animal.

**Figure supplement 3.** Mean firing rate distribution and correlation with animals' speed.

**Figure supplement 4.** Pairwise cross-correlogram (CCG) between CA1 pyramidal cells.

**Figure supplement 5.** Population synchrony remains after removing the later spikes in bursts.

In novel exploration, mice moved at a speed of 8.3±0.9 cm/s during locomotion (speed >3 cm/s) and spent 53 ± 7% and 36 ± 6% of their time in locomotion and immobility (speed <1 cm/s), respectively (n=5 mice). Meanwhile, they completed 10±2.5 laps during the 180 s imaging periods (n=5 mice).

For voltage imaging, we expressed the voltage indicator Voltron2 (*Abdelfattah et al., 2023*) in CA1 pyramidal cells (CA1PCs) that were born on embryonic day (E) 14.5, allowing us to measure both suprathreshold and subthreshold membrane potentials from multiple CA1PCs (14±4 cells imaged per field of view (mean ±s.d.) and, in total, 71 cells imaged from 5 fields of view in 5 mice; *Figure 1B*, *Figure 1—figure supplement 1A and B*). Simultaneously, we monitored the animals' positions and speed within the track while recording the local field potential (LFP) from the contralateral side of the hippocampus (*Figure 1B*, *Figure 1—figure supplement 2*). Consistent with previous studies, neurons labeled on E14.5 located more on the deep side of the pyramidal layer than those labeled on E17.5 ($t_{(601)}$=22.8, p<0.0001, Student's *t*-test; *Figure 1—figure supplement 1C and D*; *Angevine Jr, 1965*; *Bayer, 1980*; *Cavalieri et al., 2021*; *Cembrowski et al., 2016*; *Huszár et al., 2022*).

The mean firing rates of cells ranged from 2.3 to 4.3 Hz ($25^{th}$-$75^{th}$ percentiles) with a median of 3.2 Hz (n=71 cells, *Figure 1—figure supplement 3A*). Additionally, most CA1PCs exhibited elevated mean firing rates during immobility relative to locomotion, and their firing rates showed a negative correlation with the animals' speed (*Figure 1—figure supplement 3B and C*), consistent with previous research (*Adam et al., 2019*).

## Synchronous ensembles among many CA1 pyramidal neurons

*Figure 1C* shows the fluorescence traces of all neurons in a session as an example. Instances of synchronous activity, where many neurons emitted spikes within a narrow time window, were frequently observed (*Figure 1C*). To investigate synchronous activity within the neuronal population, we selected a reference cell and counted the number of spikes across all other cells that were aligned with the spikes of the reference cell. This approach yielded a grand average cross-correlogram (CCG) capable of elucidating the timing of spikes from the reference cell relative to the combined spikes of the rest of the population. Notably, the grand average CCG exhibited heightened spike counts around zero lags, unlike flattened counts calculated from randomly selected cells of different sessions (*Figure 1D and E*). The full width at half maximum (FWHM) of the peak in the grand average CCG measured 23±14ms (mean ±s.d., n=71 cells from 5 sessions; *Figure 1F*), indicating the timescale of the population synchrony. Furthermore, pairwise CCGs of many cell pairs showed significant peaks compared to jittered data (percentage of significant pairs: 63% (median), n=497 pairs, *Figure 1—figure supplement 4A and B*). Among pairs with significant peaks, the absolute values of the peak lags ranged from 0.8 to 4.3ms ($25^{th}$-$75^{th}$ percentiles) with a median of 2.1ms, and the FWHM of the peaks ranged from 20 to 46ms ($25^{th}$-$75^{th}$ percentiles) with a median of 30ms (n=315 pairs, *Figure 1—figure supplement 4C and D*). These results highlight the synchronous spiking between neurons on the millisecond timescale.

To systematically reveal individual synchronous events, we counted the number of spikes from all simultaneously recorded neurons within sliding windows of 25ms. To compare the summed spikes

with controls, we randomly jittered the spike trains to perturb spike timings while maintaining spike rates. We identified moments of population synchrony when spike sums exceeded the average of the jittered spike sums plus four standard deviations (*Figure 1C*). Using the same criterion for event detection, we observed over a 7.8-fold increase in synchronous events in the original data compared to jittered controls (mean event rate: 0.94±0.17 Hz for the original data, 0.12±0.01 Hz for the jittered data, n=10 segments, p=0.002, *W*=55, Wilcoxon signed-rank test; *Figure 1G*). The observed population synchrony was not attributable to spikes in complex bursts, as synchronous event rates did not differ significantly with or without the inclusion of later spikes in bursts (*Figure 1—figure supplement 5*). Furthermore, we calculated the percentage of neurons participating in these synchronous events to examine the synchronous ensemble sizes. The ensemble sizes ranged from 32% to 53% (25th-75th percentiles) with a median of 42%, which were larger than the synchronous ensemble sizes associated with ripples (*Buzsáki and da Silva, 2012*; *Figure 1H*). These results demonstrate synchronous ensembles of CA1PCs engaging many neurons.

## Synchronous ensembles during locomotion and immobility of novel exploration

The hippocampus exhibits state-dependent activities when rodents are engaged in locomotion and immobility (*Adam et al., 2019*; *Klausberger et al., 2003*). In novel exploration, CA1PCs showed synchronous spiking during locomotion and immobility (*Figure 2A*). To investigate whether these synchronous activities differed between these behavioral states, we compared grant average CCGs and synchronous ensembles under these conditions. During both immobility and locomotion, grant average CCGs showed prominent peaks compared to jittered data (*Figure 2B*). Moreover, synchronous ensembles were rich in both immobility and locomotion periods (*Figure 2A, B and D*). Nevertheless, the FWHMs of grand average CCGs were broader in the immobility group compared to the locomotion group (*Figure 2B and C*). Synchronous events occurred more frequently during immobility compared to locomotion (*Figure 2D*). Additionally, ensemble sizes were significantly larger in the immobility group compared to the locomotion group (*Figure 2E*). Taken together, with differences in sizes and kinetics, there were synchronous ensembles during both immobility and locomotion periods of novel exploration.

## Synchronous ensembles occur outside c-ripple episodes

CA1PCs exhibit synchronous activities when the hippocampal network emits ripple waves (*Buzsáki, 2015*; *Buzsáki and da Silva, 2012*). To test whether there is a co-occurrence of synchronous ensembles and c-ripples, we first detected c-ripples from LFP recordings. C-ripples were detected in three of the five recording sessions, and all occurred during immobility (mean c-ripple rate: 0.04±0.02 Hz during immobility, 0 Hz during locomotion). In the two recording sessions where we did not detect any c-ripples, we verified the recording quality of the LFP signal by running multiple tests after the recording sessions. C-ripples were identified in the same animals during quality tests, indicating that the absence of c-ripples during novel immobility in the recording session was not due to the deterioration of LFP recordings (*Figure 3—figure supplement 1*).

To visualize both c-ripples and synchronous ensembles, we plotted voltage traces of neurons alongside simultaneously recorded c-ripple oscillations. It was typical for synchronous ensembles to occur far away from c-ripple episodes (*Figure 3A*). To delve into the relationship between neuronal activities and c-ripples, we aligned traces of c-ripple power with population synchrony and c-ripples. In a recording session where 64 synchronous ensembles and 12 c-ripples were detected during immobility, the LFP power at c-ripple frequency (120–240 Hz) exhibited no discernible peaks within any of the synchronous events (*Figure 3B*, left upper and middle panels). Nevertheless, the same LFP power displayed substantial peaks during every c-ripple event (*Figure 3B*, right upper and middle panels). A striking contrast between synchronous events and c-ripples was also evident when spiking probabilities during these events were examined. During population synchrony, the mean spiking probability averaged across all cells was high, while the spiking probability was almost zero during c-ripple events (*Figure 3B*, lower panel). In the three sessions where c-ripples were identified, we calculated the percentages of c-ripple events coinciding with synchronous ensembles and vice versa. Remarkably, there was no co-occurrence of c-ripples and synchronous ensembles (*Figure 3C*). Furthermore, we quantified c-ripple-modulated spiking by calculating the ratio of the difference in mean firing rates

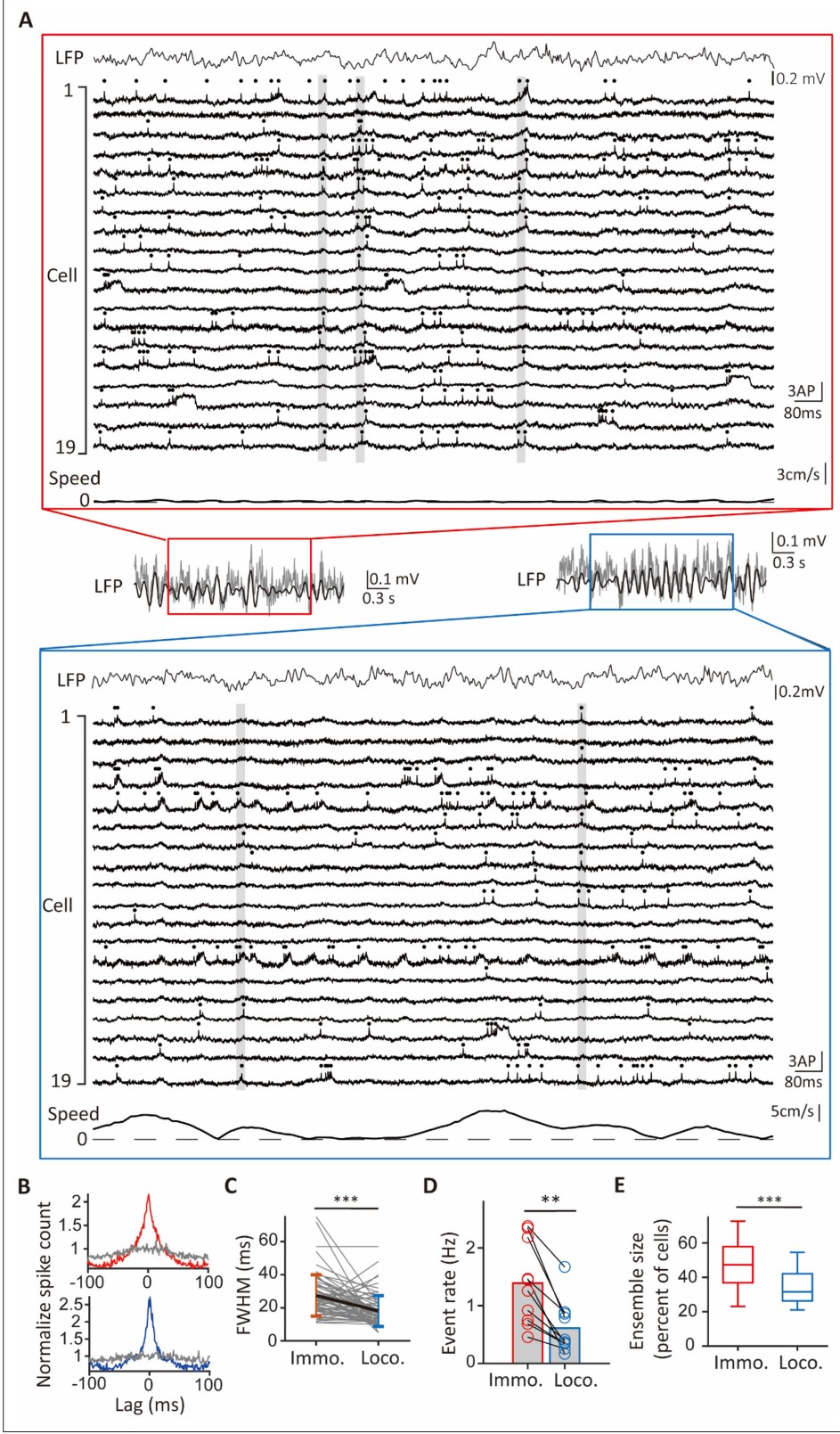

**Figure 2.** Comparison of synchronous ensembles between immobility and locomotion. (**A**) Example traces of cells in a session with identified synchronous ensembles (gray vertical bars) during periods of immobility (red rectangular frame) and locomotion (blue rectangular frame). Spikes are indicated by dots at the top of individual traces. Top: the simultaneously recorded LFP trace. Bottom: simultaneously recorded speed of the animal. Inset:

*Figure 2 continued on next page*

*Figure 2 continued*

the band-pass filtered LFP trace at the theta frequency range (black line) overlaid on the raw LFP trace (gray line). (**B**) The grand average cross-correlograms (CCGs) during immobility (red) and locomotion (blue) averaged across all cells (n=71). The gray lines represent the mean grand average CCGs calculated from jittered data. (**C**) Pairwise comparison of peak widths (FWHM) in the grand average CCGs between immobility (red) and locomotion (blue). Vertical bars represent means and standard deviations for both groups (FWHM: 27±12ms for immobility, 18±9ms for locomotion, mean ±s.d., $t_{(70)}$=5.45, p<0.001, paired t-test, n=71 cells). ***p<0.001 (**D**) Pairwise comparison of event rates of population synchrony during immobility and locomotion. Bar heights indicate group means (immobility: 1.4±0.7 Hz, locomotion: 0.6±0.5 Hz, mean ±s.d., n=10 segments, *W*=55, p=0.002, Wilcoxon signed-rank test). **p<0.01. (**E**) Boxplot of ensemble sizes for synchronous ensembles occurring during immobility (red) and locomotion (blue) (median ensemble size: 47% for immobility, n=446 events, 32% for locomotion, n=313 events, $t_{(757)}$=13.54, p<0.001, Student's t-test). ***p<0.001.

during in-c-ripple and out-c-ripple periods to the sum of the mean firing rates during both periods. Indexes of most cells were negative, and a striking 85% of cells displayed c-ripple modulation indexes of −1, indicating complete suppression of their spiking activities during c-ripples (*Figure 3D*). Taken together, during novel exploration, synchronous ensembles involved many neurons with millisecond-synchronized spikes and occurred outside the time frames of c-ripples.

It was puzzling that these CA1PCs exhibited robust spiking activities outside of c-ripples, yet generated few spikes during c-ripples. To further investigate neuronal activities during c-ripples, we established a recording condition that allowed us to capture more c-ripple episodes. Specifically, we immobilized mice in a tube to promote behaviors favoring c-ripple generation. The mice were habituated to head fixation in a tube in a room distinct from the one where imaging experiments were conducted. On the imaging day, the mice were introduced to the recording room and head-fixed under the microscope for the first time.

CA1PCs were labeled in utero on embryonic day (E) 14.5 (n=56 cells from 3 sessions in 3 mice) and E17.5 (n=55 cells from 3 sessions in 3 mice) and imaged in adult brains. Both neuronal populations exhibited prominent peaks in their grand average CCGs and significantly higher synchronous event rates compared to jittered data (*Figure 3—figure supplement 2A*, B). Approximately 40% of the recorded neurons participated in synchronous ensembles, indicating robust synchronous activity involving a substantial proportion of the recorded cells (*Figure 3—figure supplement 2C*).

In total, 1052 synchronous ensembles and 174 c-ripple episodes were detected across these imaging sessions. Consistent with findings from walking animals, few synchronous ensembles occurred during c-ripples when animals were immobilized in a tube (*Figure 3—figure supplement 3A and B*). Moreover, no distinguishable c-ripple oscillations were observed in synchronous events, and the average firing rates during c-ripple episodes were near zero (*Figure 3—figure supplement 3C and D*). At the single-cell level, 90% of neurons showed significant negative spiking modulation during c-ripples, with c-ripple modulation indexes close to −1, indicating strong suppression of spiking (*Figure 4Ai*). This suppression extended to subthreshold membrane potentials, as nearly all cells exhibited decreased fluorescence during c-ripples compared to baseline (*Figure 4Bi and Ci*). These results demonstrate that spiking activity and subthreshold membrane potentials are robustly suppressed during c-ripples.

Contextual novelty plays a critical role in shaping hippocampal neuronal activities (*Wilson and McNaughton, 1993*; *Frank et al., 2004*; *Leutgeb et al., 2004*; *Nitz and McNaughton, 2004*; *Karlsson and Frank, 2008*; *Jeewajee et al., 2008*; *Grosmark and Buzsáki, 2016*). To assess its influence, we trained mice to become familiar with the imaging procedure and the recording environment over five days and recorded CA1PC activities on the final day. Mean firing rates were significantly reduced in the familiar group compared to the novel group (Familiar group: 1.1–5.2 Hz (25th-75th percentiles), median = 2.3 Hz, n=66 cells, 6 sessions, 4 mice; Novel group: 1.7–6.0 Hz (25th-75th percentiles), median = 4.2 Hz, n=111 cells, 6 sessions, 6 mice, p=0.0083, Wilcoxon signed-rank test). Additionally, 15% of the neurons in the familiar group exhibited significantly positive spiking modulation by c-ripples, while fewer cells showed negative modulation compared to the novel group (*Figure 4A*). During c-ripples, neurons in the novel group predominantly displayed hyperpolarizing membrane voltage responses, whereas a subset of neurons in the familiar group exhibited prominent depolarizing responses (*Figure 4B*). The mean fluorescence changes in the familiar group shifted toward depolarization compared to the novel group (*Figure 4C*). Finally, synchronous event frequencies were significantly lower in the familiar context, indicating weaker synchronous activities under familiar

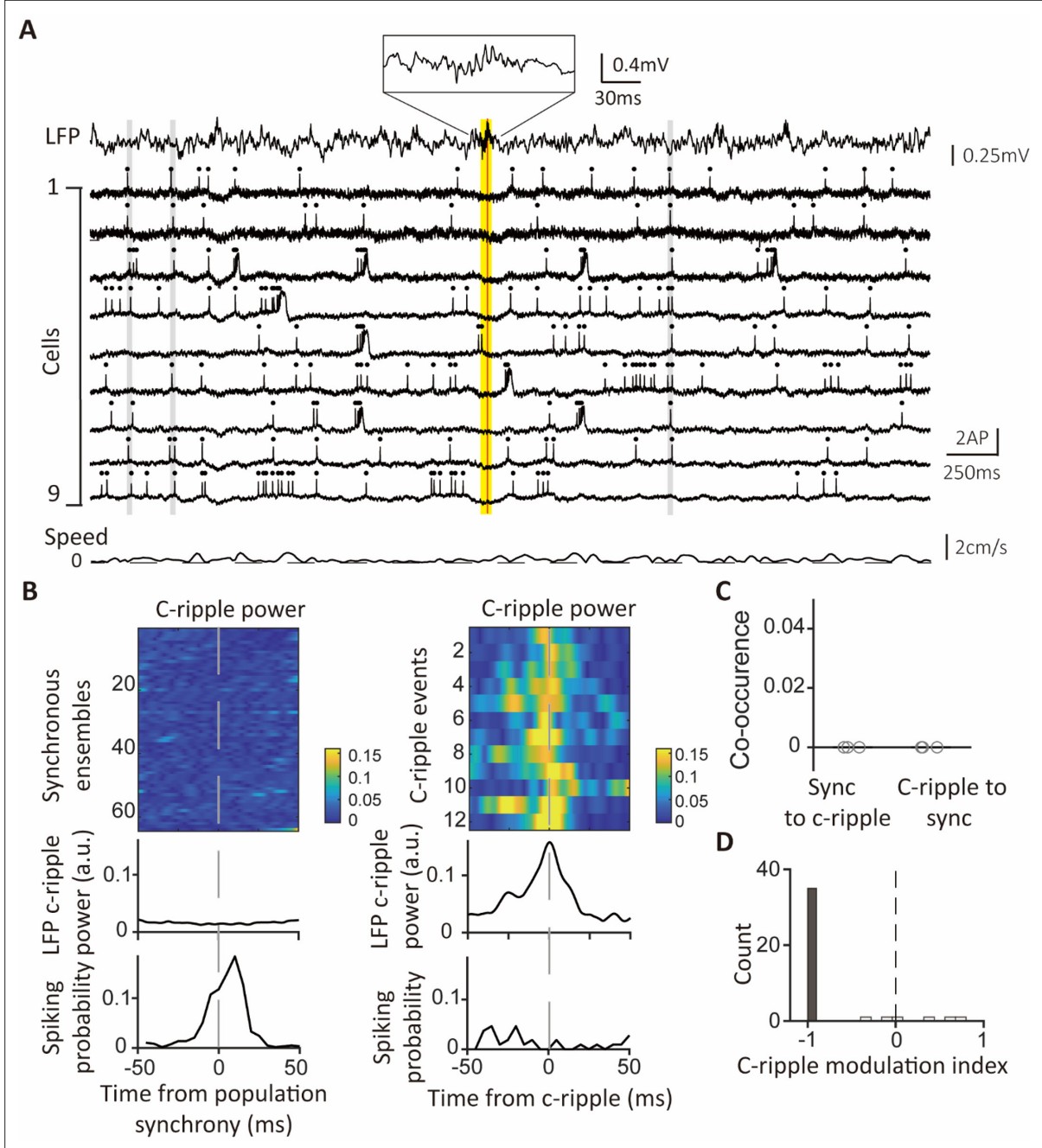

**Figure 3.** Synchronous ensembles occur outside c-ripple episodes. (**A**) Example traces with c-ripples and synchronous ensembles. Top: simultaneously recorded LFP displaying an episode of c-ripple oscillation. A red vertical line indicates the timing of the peak c-ripple power in the LFP trace. A yellow vertical bar marks the duration of the c-ripple episode. Inset at the top: a magnified view of the c-ripple oscillation. Middle: fluorescence traces of 9 CA1PCs. Gray vertical bars mark the timings of the synchronous ensembles on all traces. Bottom: animals' speed recorded simultaneously with voltage imaging. (**B**) LFP c-ripple power and spiking probability aligned to synchrony ensembles and c-ripple events. Upper panel: color-coded LFP power at c-ripple frequency (120–240 Hz) aligned to the timings of 64 synchronous ensembles (left) and 12 c-ripple events (right). Middle panel: Average LFP power of c-ripple frequency aligned to the timings of the synchronous ensembles (left) and the c-ripple events (right). Lower panel: Average spike counts of all cells aligned to the timings of the synchronous ensembles (left) and c-ripple events (right). (**C**) Percentages of co-occurrences between population synchrony and c-ripples. The co-occurrences in the 'sync to c-ripples' group represent the percentages of c-ripple episodes co-occurring with synchronous ensembles in the same episodes. The co-occurrences in the 'c-ripples to sync' group indicate the percentages of synchronous ensembles co-occurring with c-ripple episodes. Each gray circle represents an individual session. (**D**) Distribution of the c-ripple modulation indexes (n=41 cells). Gray bar: modulation indexes with significant p values (n=35 cells); white bars: modulation indexes without significance (n=6 cells).

The online version of this article includes the following source data and figure supplement(s) for figure 3:

*Figure 3 continued on next page*

*Figure 3 continued*

**Source data 1.** Excel file containing numerical data used to generate *Figure 3B*.

**Figure supplement 1.** LFP c-ripples recorded in the first and subsequent sessions in a novel maze.

**Figure supplement 2.** Population synchrony during novel immobility.

**Figure supplement 3.** Population synchrony outside c-ripple episodes during novel immobility.

conditions (*Figure 4D*). These results demonstrate that hippocampal neuronal activities, particularly synchronous ensembles, are strongly influenced by contextual novelty.

## Synchronous ensembles are associated with theta oscillations

To compare the network dynamics associated with synchronous ensembles during immobility and locomotion, we aligned LFP traces triggered to the timings of synchronous ensembles. The average LFP traces of a representative session showed prominent theta oscillations around the timings of both immobility and locomotion population synchrony (*Figure 5A*). Spectral analysis of the mean synchrony-triggered LFP traces during immobility and locomotion revealed theta frequency components of 4–12 Hz, with a slightly higher peak frequency in locomotion LFP compared to immobility LFP (*Figure 5B*). Furthermore, we computed LFP theta modulation and identified preferred phases of synchronous ensembles. In line with the previous result, synchronous ensembles displayed a high level of modulation by LFP theta oscillation (modulation strength: 0.61±0.05 during immobility and 0.74±0.04 during locomotion, n=5 sessions). Although the preferred phases varied from session to session due to differences in recording sites along the proximal-distal axis of the hippocampus, the timings of individual ensembles were consistently locked to the preferred phase of each session (*Figure 5C*). These results establish a strong link between synchronous ensembles and LFP theta oscillation.

Subthreshold membrane voltage (subVm) is one of the critical parameters supporting the generation of action potentials. To investigate the subthreshold signals underlying population synchrony, we aligned subVm traces with the timings of synchronous ensembles and observed that both immobility and locomotion synchrony rode on a subVm waveform oscillating at theta frequency (*Figure 5D*). To elucidate further the relationship between subVm theta oscillation and spikes involved in population synchrony, we calculated the subVm theta modulation and preferred phases of spikes. Spikes participating in both immobility and locomotion synchrony were strongly coupled to the subVm theta oscillation of neurons, with the mean preferred phase being in the latter half of the rising phase closer to subVm theta peaks (*Figure 5E*). Conversely, spikes not participating in synchronous ensembles had weaker modulation strength compared to spikes involved in synchrony (*Figure 5—figure supplement 1A and B*). Furthermore, the membrane voltages triggered by these two categories of spikes showed a significant difference in the hyperpolarization phase of the theta oscillation. Synchronous spikes were situated within cycles of theta oscillations featuring more hyperpolarized troughs than spikes outside synchronous ensembles (*Figure 5—figure supplement 1C and D*). Thus, compared to spikes not participating in synchronous ensembles, synchronous spikes showed a more robust modulation by theta oscillation of membrane voltage during immobility and locomotion in the context of novel exploration.

Notably, phase locking of spikes to their subVm theta oscillation and the synchronous spikes among many neurons suggest that subVms of different neurons would be correlated at the theta frequency. To explore the possibility, we divided the 180 s subVm traces into 1 s segments with 50% overlap and calculated the cross-correlation and magnitude-squared coherence function for each pair of cell segments. Segments were sorted into the immobility or locomotion group based on the animals' speed during each segment (kept below 1 cm/s for immobility and above 3 cm/s for locomotion for more than 90% of the time). In total, we analyzed 511 cell pairs from 5 sessions. Indeed, the cross-correlation of subVm between cell pairs revealed a high correlation at zero lag and periodic modulation of the correlation at theta frequency (*Figure 5F*). Furthermore, the subVm theta coherences were increased in both groups (immobility: 0.50, locomotion: 0.40, median, n=511 pairs) and were negatively correlated with soma distances (immobility: –0.46, locomotion: –0.48, n=511 pairs, *Figure 5G*). Taken together, during novel exploration, CA1PCs generate synchronous activity associated with theta oscillation and exhibit correlated and theta-coherent subVms.

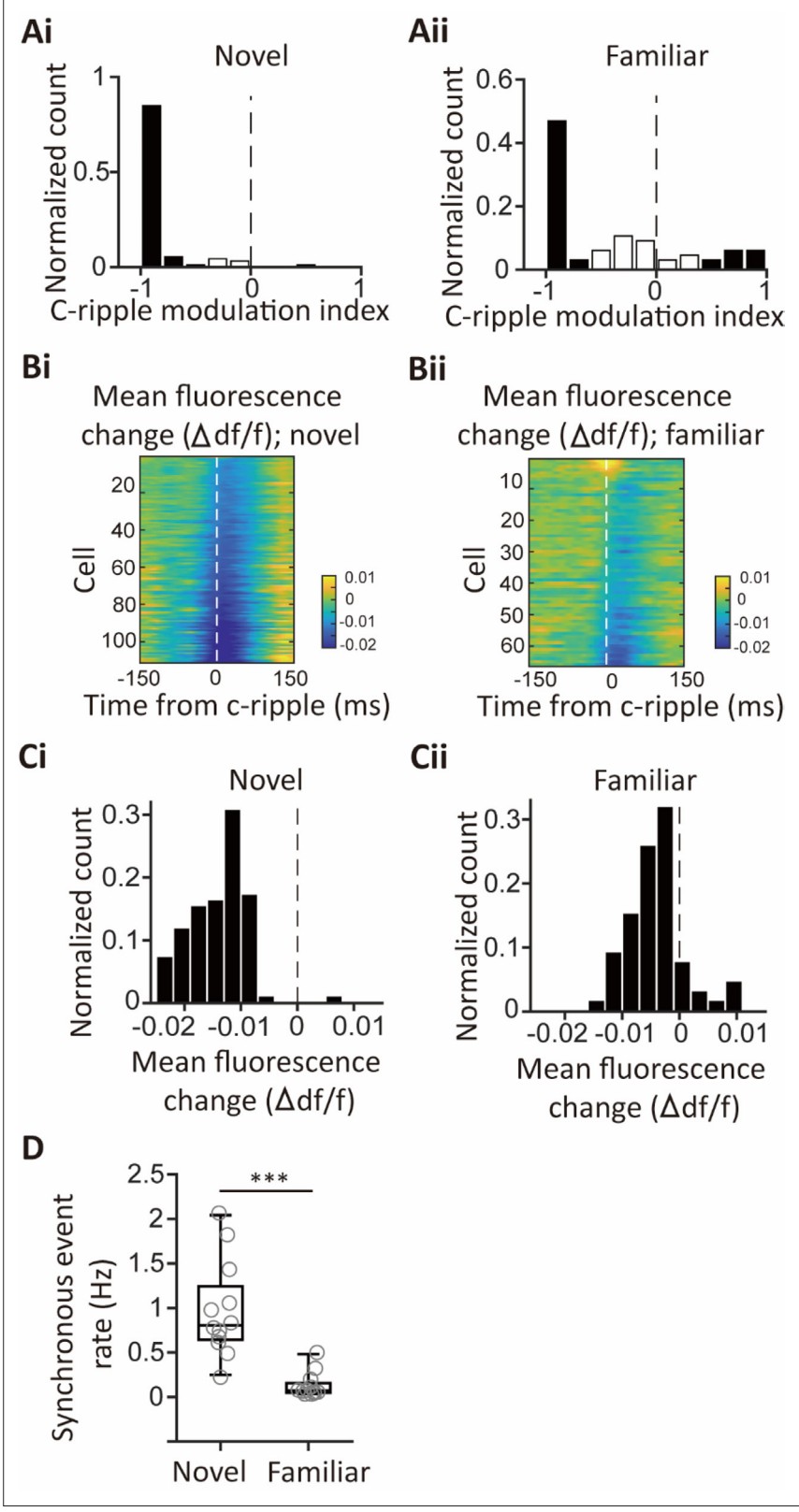

**Figure 4.** Comparison of c-ripple-associated neuronal activities and synchronous event rates between novel and familiar recording contexts. (**A**) Histograms of c-ripple modulation indices for the novel (**Ai**) and familiar (**Aii**) recording contexts. Gray bars represent modulation indices with significant *p*-values, while white bars indicate non-significant modulation indices.(**B**) Color-coded representations of mean fluorescence changes relative to baseline

*Figure 4 continued on next page*

*Figure 4 continued*

fluorescence for individual neurons, averaged across c-ripple events, in the novel (**Bi**) and familiar (**Bii**) recording contexts. (**C**) Histograms of mean fluorescence changes within a±10 ms window centered on the peak power of c-ripple events. Ci corresponds to the novel context, while Cii corresponds to the familiar context. (**D**) Boxplots of synchronous event rates for the novel and familiar recording contexts. Mean event rates were 0.98±0.16 Hz for the novel context and 0.13±0.04 Hz for the familiar context (n=12 segments for both groups, *t*(22)=5.20, p<0.001, Student's *t*-test). ***p<0.001.

Correlated intracellular theta and theta-phase locking of the synchronous ensembles raise the question of whether population synchrony among CA1PCs extends beyond synchrony derived from these effects. To address this, we analyzed population synchrony after randomizing the theta cycles during which neurons spiked, while keeping their theta phases unchanged. *Figure 5—figure supplement 2* illustrates a significant reduction in synchronous event rates following theta cycle randomization. The finding indicates spiking at specific theta cycles plays a major role in driving population synchrony.

To further investigate synchronous ensembles across different datasets, we analyzed publicly available hippocampal recordings 'hc-11' from the CRCNS repository (*Grosmark and Buzsáki, 2016*; *Grosmark et al., 2016*), where rats navigated novel mazes for water rewards (see Method). Using our algorithm, we identified a significant number of synchronous ensembles during the first three minutes of novel navigation. On average, the rates of synchronous events were 6.4-fold higher than those detected in jittered controls (mean event rate: 2.0±0.3 Hz for the original data vs. 0.32±0.03 Hz for jittered data, n=8 sessions, p=0.0078, *W*=36, Wilcoxon signed-rank test; *Figure 5—figure supplement 3A*). To assess whether ripple oscillations were associated with these synchronous ensembles, we analyzed ripple event rates and their relationship to population synchrony. During this period, ripple events were infrequent (mean ripple rate: 0.02±0.01, n=8 sessions), and ripple power peaked during ripple episodes but remained low at the timings of population synchrony (*Figure 5—figure supplement 3B*). Nevertheless, LFP traces aligned to population synchrony revealed prominent theta oscillations (*Figure 5—figure supplement 3C*). Synchronous ensembles were modulated by LFP theta oscillation (modulation strength: 0.30±0.04, n=8 sessions, p<0.001), and the timings of individual ensembles were consistently locked to the preferred phase of each session, suggesting a functional coupling of synchronous ensembles to theta oscillations important for information processing (*Figure 5—figure supplement 3D*).

## Synchrony between place cells with distinct place tunings

When animals explore a novel maze, spatially tuned spiking activity can form within a few minutes (*Rolotti et al., 2022*; *Sheffield et al., 2017*). However, how synchronous ensembles organize their spatially tuned activities during novel exploration remains to be determined. Synchronous ensembles, generating spikes coincidently in short intervals, might display correlated spiking at longer time windows that match behavioral timescales of exploration, potentially leading to shared place responses. On the other hand, CA1PCs are interconnected through inhibitory interneurons (*Takács et al., 2012*; *English et al., 2017*; *Maccaferri et al., 2000*). Modulation of activities in inhibitory interneurons by some place cells may influence the activities of other place cells, resulting in divergent spatial tunings (*Rolotti et al., 2022*; *Geiller et al., 2022*; *Trouche et al., 2016*).

To compare the spatial tunings of synchronous cell pairs, we first calculated the spatial tuning curves of individual neurons by computing the mean firing rates in every spatial bin. On average, 40% of cells displayed selective spatial tunings of their spiking activities and were qualified as place cells (*Go et al., 2021*; *Figure 6A*). Among place cell pairs, some showed similar spatial tunings, while others did not (*Figure 6B*). For instance, in an illustrative session depicted in *Figure 6B*, the seventh and eighth place cells had highly similar spatial tunings with a correlation coefficient as high as 0.7. In contrast, the fifth and tenth place cells exhibited vastly different spatial tunings with a correlation coefficient as low as –0.5.

We further explored the relationship between the similarity of spatial tunings and synchronization strength among pairs of place cells. We measured synchronization strength as the peak of the CCG normalized by the averaged jittered spike count within the same bin as the CCG peak. We segregated spikes based on these behavioral states to accommodate differences between locomotion and immobility and calculated the corresponding CCG and synchronization strength.

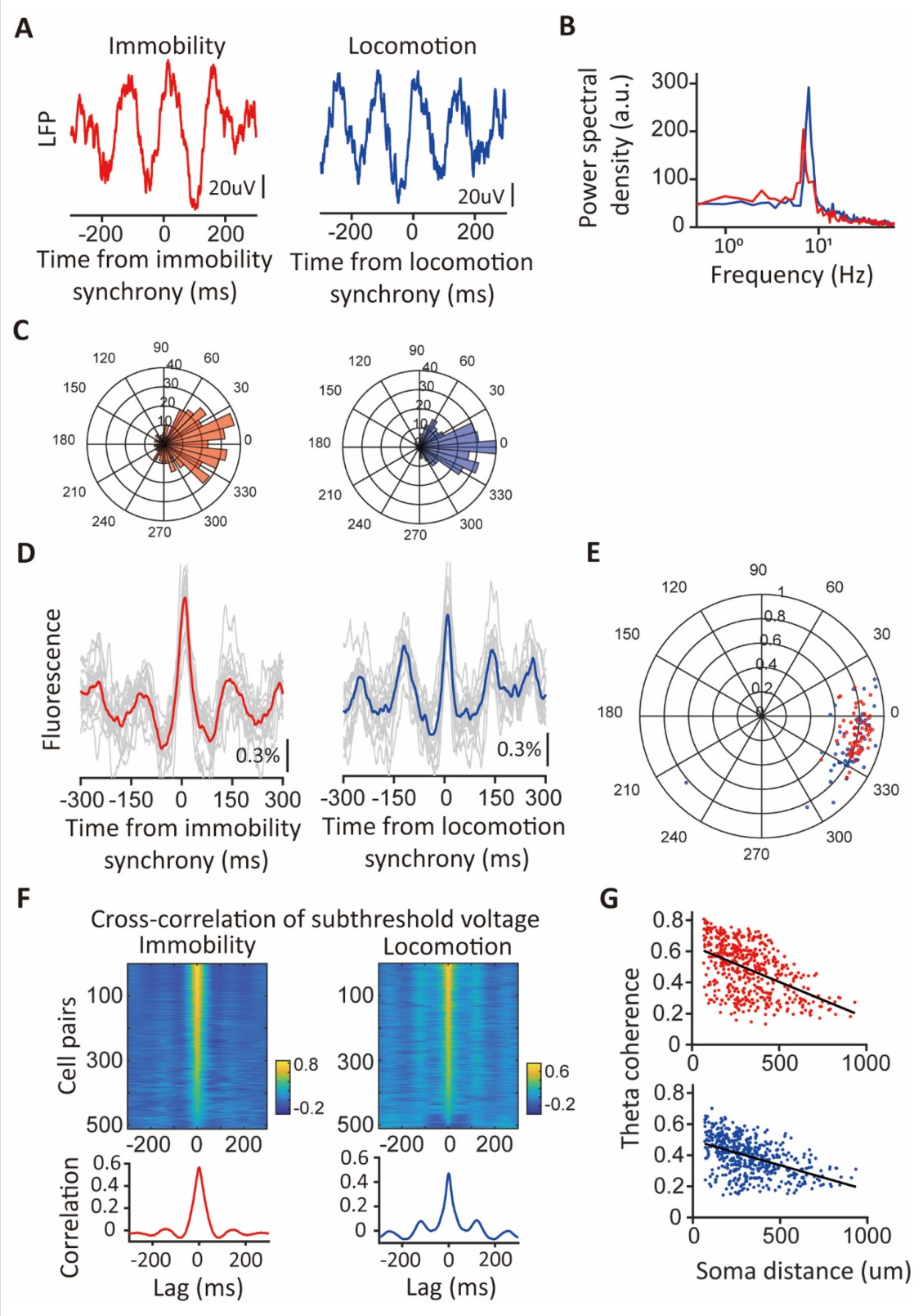

**Figure 5.** Theta oscillations in the LFP and subthreshold membrane potential were linked to population synchrony. (**A**) LFP waveforms triggered by timings of synchronous ensembles. Red: triggered by timings of immobility synchrony; blue: triggered by timings of locomotion synchrony. (**B**) Power spectral densities of the triggered LFP waveforms. Red: immobility; blue: locomotion. (**C**) Distributions of the differences between LFP theta phases of individual synchronous ensembles and their preferred phases of the session. Note that most ensembles show minor differences within ±30 degrees.

*Figure 5 continued on next page*

*Figure 5 continued*

Red: immobility; blue: locomotion. (**D**) Fluorescence traces triggered by timings of synchronous ensembles. Traces with spikes are excluded. Left panel: traces triggered by timings of immobility synchrony. Right panel: traces triggered by timings of locomotion synchrony. Thin gray lines: mean triggered fluorescence traces averaged across triggers for representative cells. Thick red and blue lines: mean triggered fluorescence traces averaged across cells. (**E**) Polar plot illustrating theta modulation of spikes participating in the immobility synchrony (red dots) and of spikes participating in the locomotion synchrony (blue dots). The angle of the dots indicates the preferred phase, and the distance from the center of the circle represents the modulation strength. Each dot represents the averages from a neuron. (**F**) Pairwise cross-correlation of the subthreshold membrane voltages (subVm). Top: color-coded cross-correlation of subVm between cell pairs during immobility and locomotion periods (n=511 pairs). Bottom: averaged cross-correlation over all cell pairs for immobility (red) and locomotion (blue) subVm. Both show a central peak at zero lag flanked by theta oscillation. (**G**) Scatter plots of theta coherence against soma distances of immobility subVm (red) and locomotion subVm (blue) between cell pairs (n=511 pairs).

The online version of this article includes the following figure supplement(s) for figure 5:

**Figure supplement 1.** SubVm theta modulation of spikes during immobility and locomotion periods of novel exploration.

**Figure supplement 2.** Significant reduction of population synchrony by randomizing spikes across theta cycles while preserving phases.

**Figure supplement 3.** Analyses of synchronous ensembles in the publicly available data.

During locomotion, we observed a robust negative correlation between synchronization strength and the similarity of spatial tunings among cell pairs. In other words, cell pairs with more precisely co-active spikes during locomotion exhibited more distinct place-tuning profiles (*Figure 6C and D*). This anti-correlation between synchronization strength and place tunings indicates coordination between place cells' temporal and spatial coding. On the other hand, synchronization strength during immobility showed little correlation with the similarity of spatial tunings among place cell pairs, suggesting that correlated activities at the millisecond timescale do not extend to slow, behavioral-relevant timescales (*Figure 6—figure supplement 1*). Thus, place cells with distinct place fields are linked by synchronous activity during novel locomotion.

## Discussion

Our findings reveal the presence of synchronous ensembles comprising a substantial number of CA1PCs during both locomotion and immobility in a novel environment. These ensembles concurrently engage neurons representing diverse aspects of the environment and are rarely observed during c-ripple events. Instead, they closely align with LFP and intracellular theta oscillations. Our data demonstrate theta-associated synchronous ensembles during spatial memory acquisition, coordinating numerous CA1 place cells that transmit information about different segments of the environment.

The correlation between observed population synchrony and theta oscillations is intriguing. Theta oscillations are typically associated with the sequential activation of neurons over time, resulting in reduced neuronal synchrony (*Mizuseki and Buzsaki, 2014*). However, theta oscillations come in different types, each characterized by distinct pharmacological properties and behavioral correlates (*Kramis et al., 1975*; *Bland, 1986*; *Buzsáki, 2002*). For example, type 1 theta, or translational theta, occurs during voluntary movements and is sensitive to N-methyl-D-aspartate receptor (NMDAR) blockers. On the other hand, type 2 theta, or attentional theta, is slightly slower and is blocked by muscarinic receptor antagonists, emerging during states of arousal or attention, such as when entering a new environment (*Sainsbury et al., 1987*; *Sainsbury, 1998*; *Barry et al., 2012*). Consistent with these distinctions, the peak of the power spectrum density shows a distinctively slower theta frequency during immobility compared to locomotion (*Figure 5B*).

During theta oscillations, cholinergic neurons in the medial septum excite downstream CA1PCs, leading to increased firing rates and membrane potentials (*Nakajima et al., 1986*; *Markram and Segal, 1990*; *Fraser and MacVicar, 1996*). Given the widespread influence of cholinergic inputs on numerous CA1PCs, theta-coupled cholinergic inputs may drive depolarization and firing across many CA1PCs. Additionally, they could enhance the likelihood of synchronicity. Furthermore, GABAergic neurons in the medial septum, activated during theta oscillations, could activate interneurons in the CA1 region (*Buzsáki, 2002*; *Tóth et al., 1997*; *Colgin, 2013*). These CA1 interneurons, in turn, have the capability to entrain the activities of pyramidal neurons at both sub- and suprathreshold levels, thereby promoting synchronization among pyramidal neurons (*Cobb et al., 1995*; *Stark et al., 2013*).

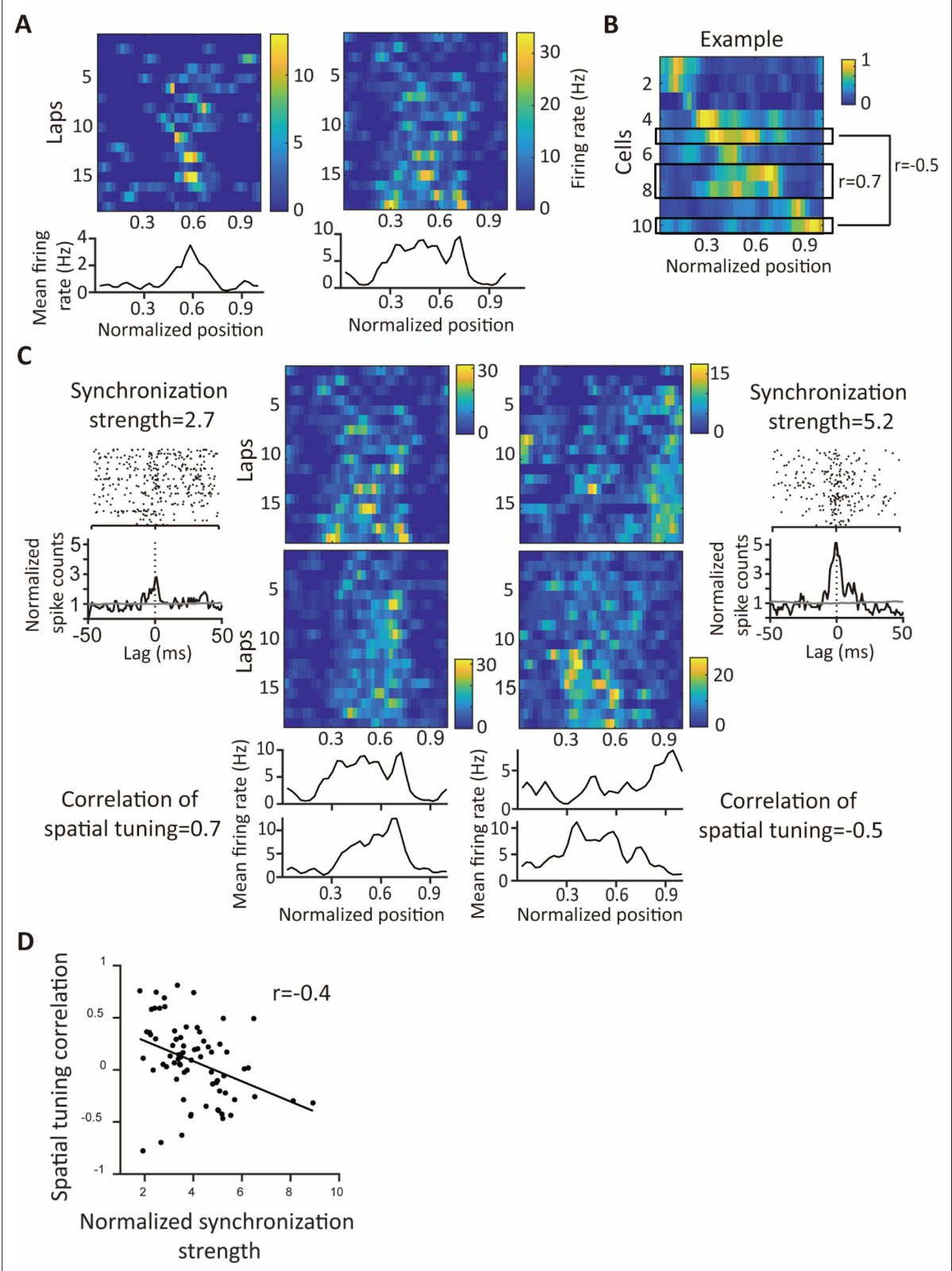

**Figure 6.** Negative correlation between spatial tuning similarity and synchronization strength during novel exploration. (**A**) Spatially binned rate maps of place cells. High firing rates are indicated in yellow, and low firing rates are noted in blue. Bottom: the mean firing rates along spatial bins averaged across laps. (**B**) Spatial tunings of 10 place cells from an example session. Cells are sorted based on the locations of their peak firing rates. (**C**) Two example pairs of neurons with cross-correlograms during locomotion and spatially binned firing rate maps. (**D**) A scatter plot of correlation coefficients

*Figure 6 continued on next page*

*Figure 6 continued*

of spatial tunings against normalized synchronization strengths for place cell pairs. Each dot represents a pair of place cells. Synchronization strengths are negatively correlated with similarities of spatial tunings (Spearman correlation coefficient = −0.4, p<0.001, n=73 pairs).

The online version of this article includes the following figure supplement(s) for figure 6:

**Figure supplement 1.** Synchronization strengths during immobility show little correlation with the similarity of spatial tunings between place cells.

In summary, cholinergic and GABAergic neuron activation during theta waves may contribute to the excitation of CA1PCs and the consequent increase in their synchronicity.

Theta oscillations have long been implicated in memory acquisition. Specifically, theta power correlates positively with learning performance (*Landfield et al., 1972*; *Berry and Thompson, 1978*; *Belchior et al., 2014*). Perturbing generation of theta waves leads to impairment in memory acquisition (*Winson, 1978*; *Mitchell et al., 1982*; *Mizumori et al., 1990*; *M'Harzi and Jarrard, 1992*; *Wang et al., 2015*), while electrical stimulation restoring theta rhythms rescues learning performance (*McNaughton et al., 2006*; *Lee et al., 2013*). Furthermore, increased spike-theta coherence correlates with better memory, and stimulations that boost theta phase-locking of spikes enhance memory (*Hasselmo et al., 2002*; *Rutishauser et al., 2010*; *Siegle and Wilson, 2014*). Despite abundant evidence supporting a link between theta oscillations and memory, the precise network mechanism by which theta oscillations support memory acquisition remains unclear. Theta oscillations are hypothesized to link spatially distributed neurons into functional ensembles to support memory acquisition (*Colgin, 2016*). The identified synchronous ensembles support the hypothesis, providing a network substrate that links theta rhythms to the initial stage of memory acquisition.

Recent studies have demonstrated elevated coactivity of hippocampal principal cells during theta-associated dentate spikes, events that are temporally coupled to theta oscillations and distinct from sharp-wave ripples (*McHugh et al., 2024*; *Dvorak et al., 2021*; *Farrell et al., 2024*). These findings highlight a network state characterized by rich synchronous ensembles associated with theta, but not ripple, oscillations—a pattern consistent with our results. Our analyses of publicly available data further support these observations, revealing theta-associated population synchrony in a different dataset (*Figure 5—figure supplement 3*). These results suggest that hippocampal neurons achieve synchrony through network mechanisms distinct from ripple-related events, reinforcing the hypothesis that this phenomenon is generalizable across experimental conditions and behavioral contexts.

Previous studies have reported that a small percentage of CA1PCs generate spikes during ripple episodes (*Buzsáki, 2015*; *Buzsáki and da Silva, 2012*; *Yagi et al., 2023*). These measurements are typically conducted in familiar environments. In contrast, when animals are introduced to a novel environment, c-ripple-associated spiking activities are strongly suppressed, and CA1PCs exhibit prominent hyperpolarizing responses in their membrane potentials (*Figures 3 and 4Ai-Ci* and *Figure 3—figure supplement 3*). Interestingly, after familiarization training, a larger proportion of CA1PCs display positive c-ripple modulation of spiking activities and depolarizing responses associated with c-ripples (*Figure 4Aii–Cii*). Consistent with the observation of ripple-associated spiking activity in familiar environments, our study underscores the influence of contextual novelty on hippocampal dynamics. Specifically, brain states associated with novelty may modulate the activity of neurons upstream of CA1PCs, driving their membrane potentials toward hyperpolarization during c-ripples. While the reduced correlation between c-ripples and spikes in novel environments may initially appear counterintuitive, this unique neuronal activity pattern likely reflects adaptive changes in hippocampal circuits related to the animals' behavioral states, such as heightened vigilance in novel environments.

While contralateral LFP recordings can capture large-scale hippocampal theta and ripple oscillations, they do not fully reflect ipsilateral-specific dynamics, such as variations in theta phase alignment or locally generated ripple events (*Buzsáki et al., 2003*; *Szabo et al., 2022*; *Huang et al., 2024*). Given that ripple oscillations can emerge locally and independently in each hemisphere, interpretations based on contralateral recordings must be made with caution. Further studies incorporating simultaneous ipsilateral field potential recordings will be essential to more precisely understand local-global network interactions. Despite these considerations, our findings provide robust evidence for the existence of synchronous neuronal ensembles and their role in coordinating newly formed place cells. These results advance our understanding of how synchronous neuronal ensembles contribute to spatial memory acquisition and hippocampal network coordination.

Although CA1PCs are known to exhibit weaker recurrent connectivity than that of CA3 pyramidal neurons, CA1PCs did communicate through monosynaptic connections (*Deuchars and Thomson, 1996*). These connections can be dynamically adjusted by the spike-timing-dependent plasticity mechanism (*Caporale and Dan, 2008*; *Markram et al., 1997*; *Bi and Poo, 1998*), and therefore could be affected by the transient population synchrony. In addition, CA1PCs are known to communicate through inhibitory interneurons. Multiple pyramidal neurons converge onto a single interneuron, and individual connections are often weak and unreliable (*English et al., 2017*; *Ali et al., 1998*; *Ali et al., 1998*). This connectivity scheme requires multiple pyramidal neurons spiking synchronously within the time window of synaptic integration to effectively activate postsynaptic neurons (*English et al., 2017*). Furthermore, population synchrony among CA1PCs may propagate to downstream areas, synchronizing neurons outside the hippocampus. With their large ensemble sizes, these synchronous ensembles may synergistically convey relevant information or reorganize the connectivity of the neuronal networks.

Prior studies in sensory systems have commonly revealed that neurons with synchronous activities typically share similar sensory responses (*Alonso et al., 1996*; *Engel et al., 1991*). However, the synchronous ensembles we have identified often encompass neurons with distinct preferences for place fields. Pairwise synchrony demonstrates a negative correlation with place field overlap (*Figure 6*). These findings indicate that synchrony does not necessarily imply similar tunings between neurons. Furthermore, these synchronous ensembles involve place cells encoding different spatial information. Recognizing the crucial role of neuronal synchrony in binding distributed neurons in the brain (*Axmacher et al., 2006*; *Wang, 2010*), population synchrony may serve as a mechanism to integrate diverse features of the same environment into a cohesive mental map.

## Methods

### Animal

Time-pregnant wild-type C57BL/6 J female mice underwent in utero virus injection of CamkII-cre virus (105558-AAV1, Addgene) into the left side of the lateral ventricles. The resulting offspring were raised in a breeding room under controlled temperature and humidity conditions and a 12 hr light/dark cycle (lights on from 7:00 to 19:00). The mice used in the experiments were between 12 and 18 weeks old at the time of the recordings. The present study followed institutional guidelines for animal treatment and complied with relevant legislation from the IACUC of National Yang Ming Chiao Tung University. All surgical procedures, behavioral training, and recording protocols were approved by the IACUC of National Yang Ming Chiao Tung University (IACUC Approval No: 1100427).

### In utero virus injection

Time-pregnant mice on embryonic day 14.5 or 17.5 were anesthetized with isoflurane (induction:4–5%; maintenance:1–3%) before surgery. The animals were placed on a heating pad in a supine position to maintain body temperature. Aseptic procedures were implemented to maintain sterile conditions during the surgery. Uterine horns were exposed one at a time using spoons and fingers. Warm and sterile phosphate-buffered saline (PBS) was used to rinse the uterus to prevent it from drying out. Micropipettes loaded with CamkII-cre virus solution were used to inject 0.2 µL of virus suspension into the left side of the lateral ventricle of each embryo. The injection resulted in Cre expression among neurons born on the day of injection, with earlier injection labeling neurons located on the deeper side of the cell layer. The uterine horn was then gently returned to the abdominal cavity. The exact process was repeated to the other uterine horn. After the surgery, ketorolac (6 mg/kg) and cefazolin (1 g/kg) were administered for two days to minimize pain and inflammation.

### Cranial window and cannula implantation

The offspring of the mice that had undergone in-utero injection were subjected to chronic hippocampal window surgery when they were between 2 and 4 months old. The animals were initially anesthetized using isoflurane (induction: 4–5%; maintenance: 1–3%). Topical anesthesia was applied by administering 0.5% lidocaine to the wound margins. After achieving deep anesthesia, an incision was made in the skin, and a circular section of the skull was removed, centered 2 mm caudal and 1.8 mm left to the bregma. Using 30-gauge needles and forceps, the dura was removed from the

exposed area, and cortical tissue within the craniotomy was aspirated. A serial dilution experiment was conducted to determine an optimal titer of the virus carrying Voltron2 genes, minimizing cell toxicity, for use in this and in previous imaging experiments (*Abdelfattah et al., 2023*). A fine injection pipette (tip diameter 10–60 µm) was used to inject AAV2/1-CAG-flex-Voltron2-ST ($2.7 \times 10^{12}$ GC/ml, a generous gift from Dr. Eric Schreiter and the GENIE team at HHMI Janelia Research Campus) into the exposed regions at a depth of 200 µm (up to six injection sites and 100–200 nL of viral suspension). An optical chamber was constructed by placing a cannula with a cover slip attached at the bottom over the craniotomy and sealing it with dental acrylic or C&B Metabond (Sun Medical). A custom-built titanium frame was then cemented to the animal's head using dental acrylic or C&B Metabond. Mice received ketorolac (6 mg/kg) and cefazolin (1 g/kg) for 2 days following surgery to minimize pain and inflammation.

## Behavioral training

After at least two weeks of recovery from the window surgery, the mice underwent behavioral training to ensure they were calm and attentive in the test environment. For mice subjected to imaging experiments where they explored an air-lifted plastic track, the initial training phase involved acclimating the mice to head fixation and treadmill running, which took place over 3–5 days in a separate room distinct from the recording setup. Following this training, the imaging experiments were conducted. On the day designated for imaging, the mice were introduced to the recording room for the first time just before the imaging session. During the imaging experiments, the mice were securely held in a head-fixed position beneath the microscope and on an air-lifted plastic track that rested on an air table (AirLift, Neurotar; *Figure 1A*). The track was adorned with various shapes of signs to help the mice navigate the track. Subdued blue lighting was provided to illuminate the track. Before the initiation of imaging, the mice were confined in a corner of the track using gates. Subsequently, the gates were opened upon the start of imaging, allowing the mice to explore the new environment during their initial laps (*Figure 1A*, Top view). The Mobile HomeCage tracking system (Neurotar) was used to track the mice's position and speed. During the acquisition of image frames, TTL pulses were sent by the high-speed camera to the Mobile HomeCage tracking system to facilitate data alignment. Mice subjected to imaging experiments with head fixation in a tube were first trained in a separate room, distinct from the recording setup, for 1–2 days to habituate them to the head fixation procedure. Following this training, the mice were introduced to the recording room for the first time immediately before the imaging session. During imaging, the mice were securely positioned in a head-fixation setup within an acrylic tube and remained stationary. To further familiarize the mice with the recording context, the imaging procedure was repeated for 4 days without injecting the JF-HaloTag ligand. On the fifth day, the JF-HaloTag ligand was injected prior to imaging, and voltage imaging was performed following the same procedure as in the previous sessions.

## Voltage imaging

$JF_{552}$-HaloTag ligand (*Zheng et al., 2019*) (a generous gift from Dr. Luke Lavis) was first dissolved in DMSO (20 µl, Sigma) and then diluted in Pluronic F-127 (20 µl, P3000MP, Invitrogen) and PBS to achieve a final concentration of 0.83 mM of $JF_{552}$-HaloTag ligand. The solution was then injected intravenously through the retro-orbital sinus. Imaging sessions were initiated 3 hr after the injection of the $JF_{552}$-HaloTag ligand. Fluorescence sensors were excited using a 532 nm laser (Opus 532, Laser Quantum) with an excitation filter (FF02-520-28, Semrock), and the emitted light was separated from the excitation light using a dichroic mirror (540lpxr Chroma) and passed through an emission filter (FF01-596183 Semrock). A 16 X, 0.8 NA objective (Nikon) was used to collect the emission light which was then imaged on a CMOS camera (DaVinci-1K, RedShirt) with a 50 mm camera lens (Nikkor 50mm f1.2, Nikon) as the tube lens. Images were acquired using Turbo-SM64 (Sci-Measure) at 2 kHz with a resolution of 190x160 pixels, corresponding to a field of view of 1.4x1.2 mm. The number of frames in an image session was set at 360,000, corresponding to a duration of 180 s per session. Imaging depths ranged between 70 and 170 µm from the window surface, where the CA1 pyramidal neurons were located. Time-lapse images were collected, and image stacks of the same field of view along the z-axis were acquired. The image stacks consisted of 100 planes separated by two µm, covering the depth from the window surface to 200 µm from the surface.

## LFP recording

The LFP signal from the contralateral CA1 region of the hippocampus was simultaneously recorded during voltage imaging. To implant the LFP electrode, a small craniotomy was performed at the dorsal CA1 coordinate (2 mm caudal and 1.8 mm right to the bregma). A tungsten electrode (#100211, 38 µm in diameter, polyimide insulated, California Fine Wire) was inserted into the dorsal CA1 region, and the wire and pins were cemented in place with dental acrylic. In order to target the CA1 pyramidal layer, the depth of the electrode was controlled and tracked by a micromanipulator (Sutter Instrument, MP-285). Pyramidal layer-specific signatures of the LFP signals, such as complex bursts and high-frequency ripple oscillations, were used to guide the targeting. The electrode location was verified by using tungsten electrodes coated with DiI (Invitrogen, D282), followed by brain slicing after the recordings. The electrical signal was amplified and filtered (1–1 kHz) using a DAM80 amplifier (Word Precision Instruments). The amplified signal was acquired by an I/O device (National Instruments USB-6341) and recorded using WaveSurfer (Adam Taylor, HHMI Janelia Research Campus) at 10 kHz during implantation and TurboSM64 (RedShirt) at 8 kHz during imaging.

## Imaging somata and dendrites in vivo using a two-photon microscope

Immediately after voltage imaging, Voltron-labeled neurons were examined in vivo under a two-photon microscope to assess the integrity of their membranes and dendrites. Images were acquired using a two-photon scanning microscope equipped with a femtosecond-pulsed laser (Chameleon, Coherent) and a resonant scanner controlled by ScanImage 4.2 software (*Pologruto et al., 2003*) (http://scanimage.org). A 16 x water-immersion objective (Nikon, 0.8 NA, 3 mm working distance) was used for imaging. Z-stacks of images (512x512 pixels, 600µm x 600µm) were collected with excitation light at 800 nm. The emission light was separated from the excitation light using a primary dichroic mirror (FF735-Di02−25x36, Semrock), followed by a secondary dichroic mirror (565dclp, Chroma) and an emission filter (FF01-609/57-30-D, Semrock). Each image stack consisted of 100 planes, separated by 2 µm, covering depths from the window surface to 200 um below the surface.

## Histological processing and quantification of soma depth

Mice with Voltron2$_{552}$ labeling were anesthetized with intraperitoneal injections of ketamine (250 mg/kg) and xylazine (25 mg/kg) and transcardially perfused with 4% paraformaldehyde in phosphate-buffered saline (PBS; 2 mL/g). The brains were post-fixed in 4% paraformaldehyde overnight and then washed in PBS. Coronal brain sections, 50 µm thick, were prepared using a VT1200 vibratome (Leica). Sections were mounted with Vectashield mounting medium containing DAPI (VectorLabs) to visualize nuclei. The soma depth of Voltron-labeled cells was measured relative to the border between the *stratum pyramidale* (SP) and the *stratum radiatum*. Depth values were normalized to the thickness of the SP, with larger depth ratios indicating somata located deeper along the radial axis.

## Voltage imaging preprocessing and spike detection

Fluorescence intensities were corrected for brain movement using rigid registration. Regions of interest (ROIs) were manually selected by grouping pixels that cover individual somata. Previous studies have described and validated the procedure for imaging preprocessing and spike detection (*Abdelfattah et al., 2023*; *Huang et al., 2024*; *Abdelfattah et al., 2019*). In short, the fluorescence intensities of individual neurons were calculated by averaging the fluorescence intensities of pixels from the same ROIs. Bleaching was corrected by calculating the baseline fluorescence ($F_0$) at each time point as an average of the fluorescence intensities within ±0.5 s around the time point. The dF/F was calculated as the $F_0$ minus the fluorescence intensity of the same time point divided by $F_0$. Positive fluorescence transients were detected to identify spikes from the high-passed dF/F traces created by subtracting the dF/F traces from the median-filtered version with a 5 ms window. To simulate the noise of recordings, high-passed dF/F traces were inverted, and the amplitudes of the transients detected from the inverted traces were used to construct a noise distribution of the spike amplitudes. A threshold was set by comparing the amplitudes of the detected transients with the noise distribution of the spike amplitudes to minimize the sum of type I and type II errors. Spikes were first detected when transients were larger than the threshold. Then, spike amplitudes smaller than half of the top 5% spike amplitudes were excluded. The signal-to-noise ratio (SNR) was calculated for each neuron as a ratio of the averaged spike amplitudes over the standard deviation of the high-passed dF/F traces, excluding

points 2ms before and 4ms after each detected spike to estimate the quality of the recordings. The crosstalk between ROIs was routinely checked by examining the refractory periods of neurons in auto-correlograms. Only neurons meeting the following criteria were included: (1) SNR larger than 5, (2) full width at half maximum of the spike waveform longer than 0.8ms, (3) mean spike rates higher than 0.1 Hz, (4) distances between soma pairs at least 70 μm. Sessions with at least nine neurons meeting the selection criteria were used for further analysis.

### Behavioral states of locomotion and immobility
The periods of locomotion were defined as instances when the animals' speed exceeded 3 cm/s, while periods of immobility were determined as instances when the animals' speed fell below 1 cm/s.

### Mean firing rates during locomotion and immobility
To calculate the mean firing rates during locomotion, we counted the number of spikes during loco-motion periods and divided this number by the total duration of locomotion. Similarly, for mean firing rates during immobility, we calculated the number of spikes that occurred during immobility periods and divided this number by the total duration of immobility.

### Correlation between instantaneous firing rate and animal speed
To estimate the instantaneous firing rates of individual neurons over time, we convolved their spike trains with a Gaussian window of 250ms. Subsequently, we calculated the Pearson correlation coefficient between these instantaneous firing rates and the animal's speed for each neuron.

### Grand average cross-correlogram (CCG)
Grand average CCGs were generated by constructing histograms of relative spike timings between spikes of a reference cell and spikes of all other neurons in the same session as the target cell popu-lation. Histograms were binned using 1 ms time bins. Spike counts were normalized by the number of reference spikes times the number of all other neurons. An equal number of cells from different sessions were randomly selected to form the shuffled target cell population to assess the statistical significance of the CCG peaks. Subsequently, grand average CCGs were computed using shuf-fled spike trains for comparison. Grand average CCGs were calculated using all spikes in the 180 s recording periods, subsets of spikes during immobility periods (immobility grand average CCGs), and subsets of spikes during locomotion periods (locomotion grand average CCGs).

### Detection of the population synchrony
To assess population synchrony, we counted the number of spikes from all neurons within a sliding window of 25ms. To test the significance of the synchrony, we generated surrogate data by jittering spike timings within a±75 ms window 500 times. The means and standard deviations of each sample point were calculated over the jitter population. Population synchrony was detected as peaks when the spike count in the original data exceeded the mean plus four standard deviations of the spike counts in the jittered data. Synchrony was also evaluated in modified spike trains, including those where later spikes in bursts were removed and those where spike times were shuffled across theta cycles while preserving phases. To estimate synchronous event rates, we divided the entire 180 s recordings into halves and calculated the event rate in each of the 90 s segments. Additionally, ensemble sizes were quantified by counting the percentage of neurons participating in each synchronous event.

To detect population synchrony in a publicly available dataset, we analyzed the 'hc-11' dataset from CRCNS, which contains hippocampal recordings from rats running on novel mazes for water rewards at designated locations (*Grosmark and Buzsáki, 2016*; *Grosmark et al., 2016*). Spikes recorded during the first three minutes of the novel-maze running epochs were used to identify synchronous ensembles and generate jittered control data. Spike counts for both the original and jittered data were calculated using a sliding 25 ms window. The algorithm for testing the significance of synchrony was the same as described above. Synchronous event rates were estimated using the entire 3 min spike trains for both the original and jittered data.

### Pairwise cross-correlogram (CCG) of neuronal pairs
To compute the CCG, we first determined the time difference between spikes in the target and reference neurons and then binned these values into 1 ms time bins. CCGs were only computed for

pairs with more than 100 counts within the CCG window of ±500ms to ensure reliable correlation. We defined the peak of the CCG as the local maxima within ±30ms of lag. To assess the significance of cross-correlations, we generated surrogate data by jittering the spike times of the target neuron within a±75ms window. We computed the CCG peak for a thousand iterations and calculated p-values by comparing the percentile of the peak in the null distribution of the jittered peak values. The synchronization strength was estimated as the ratio between the original CCG peak and the mean spike count of the same bin as the original CCG peak averaged across the jittered histograms. For display purposes, the CCGs were smoothed using a 5-point moving average. CCGs were calculated using all spikes in the 180 s recording periods, subsets of spikes during immobility periods (immobility pairwise CCGs), and subsets of spikes during locomotion periods (locomotion pairwise CCGs).

## LFP c-ripple analysis

To detect c-ripples, we performed band-pass filtering of the LFP signal between 120 and 240 Hz, followed by computing the envelope using the absolute values of the Hilbert transform, which was then low-pass filtered at 20 Hz. The start and end times of c-ripples were identified by setting the upper and lower thresholds at the mean plus 7 and 3.5 times the standard deviation of the envelope, respectively. For a c-ripple event to be included, three conditions had to be met: (1) values within the event were higher than the lower threshold, (2) at least one value exceeded the upper threshold, and (3) the minimum duration of the event was 30ms. The start and end times of a c-ripple event were defined as the positive and negative crossings of the lower threshold, respectively. We defined in-c-ripple periods as the time between the start and end of detected c-ripple events, while out-c-ripple periods were periods outside the in-c-ripple periods. The envelope peak values of the detected c-ripple events were used as the time of c-ripple occurrence. These timings were utilized as trigger timings to calculate averages triggered by c-ripples. Two co-occurrences were quantified: the percentages of c-ripple oscillations that occurred with synchronous ensembles and the percentages of synchronous ensembles that occurred with c-ripple oscillations. The same ripple detection algorithm was applied to the hc-11 dataset. To account for artifacts in the recordings, raw traces and power spectra of the detected events were plotted and inspected for false positives.

## C-ripple modulation index

We utilized the c-ripple modulation index to assess changes in firing rates during c-ripples compared to periods outside c-ripples. This index was computed as the ratio of the difference in mean firing rates during in- and out-c-ripple periods to the sum of the mean firing rates during both periods. Specifically, we first calculated the mean firing rate during in-c-ripple periods and the mean firing rate during out-c-ripple periods. Next, we subtracted the mean firing rate during out-c-ripple periods from the mean firing rate during in-c-ripple periods and divided the result by the sum of the mean firing rates during both periods. The resulting c-ripple modulation index indicates the magnitude and direction of the firing rate modulation during c-ripples compared to periods outside c-ripples. To assess the significance of the modulation index, we shuffled the spike timings and repeated the calculation a thousand times to generate a distribution of modulation indices from the shuffled data. The original modulation index was then compared to this distribution, and its significance was determined if it exceeded the 95% confidence level of the shuffled indices.

## LFP traces triggered by population synchrony and analysis

The midpoints of the synchronous event intervals were utilized as triggers to align LFP traces from both our recordings and the hc-11 dataset. These traces were aligned within a±1 s window around the trigger time points. To calculate the spectral density functions of the aligned LFP traces, we performed a 16,384-point fast Fourier transform of the 2 s LFP traces. The power spectra were then estimated as the squared absolute values of the Fourier coefficients. C-ripple power was calculated as the sum of the power distributed between 120 and 240 Hz. To calculate theta modulation strength and preferred angles of the synchronous ensembles, LFP was first filtered within the theta frequency range. Then, the phase $\varphi(t)$ and the instantaneous amplitude $A(t)$ of the filtered LFP were computed using the Hilbert transform. A vector $V_k = A(t_k) e^{i\varphi(t_k)}$ was assigned to each synchronous ensemble that occurred at time $t_k$. The modulation strength and the preferred phase angle were determined as

the absolute value and the phase angle of the summed vector over all ensembles, respectively. Modulation strengths were normalized by the total length of all vectors.

### Subthreshold membrane voltage (subVm)

To compute the subVm of individual neurons, we began by determining the difference between the raw dF/F signal and the high-pass version of dF/F. Next, we calculated the spike threshold for each cell by identifying the voltage value corresponding to the peak of the first derivative of the spike waveforms. Any fluorescence values in the slow dF/F time courses that surpassed the spike thresholds were linearly interpolated.

### SubVm traces triggered by population synchrony

The midpoint of the time intervals, during which synchronous ensembles were detected, served as the reference points for aligning fluorescence traces within a±300ms window around those reference points. To eliminate any interference from membrane potential fluctuations associated with spikes, we excluded aligned traces in cases where any spike occurred within the time intervals of the aligned traces.

### SubVm theta modulation and phase preference of spikes occurred inside and outside the population synchrony events

The subVm was initially filtered within the theta frequency range. Subsequently, the phase $\varphi(t)$ and the instantaneous amplitude $A(t)$ of the filtered subVm were computed using the Hilbert transform. Spikes were categorized into four groups: those occurring during or outside immobility synchronous events and those occurring during or outside locomotion synchronous events. Within each category, a vector $V_k = A(t_k) e^{i\varphi(t_k)}$ was assigned to each spike that occurred at time $t_k$. The modulation strength and preferred phase angle were determined as the absolute value and phase angle, respectively, of the summed vector over all spikes within a category. To facilitate comparison between neurons, modulation strengths were normalized by the total length of all vectors. The significance of the modulation strength was tested by shuffling the spike timings and recalculating the modulation strength a thousand times to generate a distribution based on the shuffled spike timings. The original modulation strength was then compared to the distribution, with significance determined if it exceeded the 95% confidence interval of the shuffled values. Significant modulation strengths were plotted and compared across groups.

### Spike-triggered fluorescence traces

We extracted segments of fluorescence traces using a±300ms time window centered on the spike timings. To examine variations in fluorescence waveforms triggered by spikes within and outside synchronous events, we categorized the fluorescence traces based on whether the spikes occurred within or outside these events. Subsequently, we performed pairwise comparisons of the fluorescence values from the same neuron, concentrating on spikes occurring during corresponding behavioral states. Neurons with four or fewer triggering events in any of these categories were omitted from the analysis.

### Cross-correlation and coherence analysis to the subVm traces

To calculate cross-correlation and theta coherence of subVm between pairs of neurons during locomotion and immobility, we initially divided the 180 s subVm traces into 1 s segments with a 50% overlap. These segments were then categorized into the locomotion group if the animal's speed remained above three cm/s for more than 90% of the time within that segment. Similarly, segments were sorted into the immobility group if the animal's speed remained below one cm/s for more than 90% of the time within that segment. Subsequently, we computed the cross-correlation and magnitude-squared coherence function of frequency for pairs of co-occurred segments belonging to a specific cell pair. Mean cross-correlations of individual cell pairs were averaged across segments that occurred in the same behavioral states (i.e. locomotion or immobility states). To estimate theta coherence, we averaged the values of the coherence function within the theta frequency range (4–12 Hz). Theta coherences from the same cell pair during the same behavioral state were pooled together, and the average was used as an estimate for the theta coherence of that cell pair during a particular behavioral state.

Finally, Spearman correlation coefficients were calculated between the mean theta coherences of cell pairs and their soma distances.

## Spatial analysis

We partitioned a 90-cm-long track into 36 spatial bins to analyze spatial tuning properties. For each cell, the spatial tuning curve was computed by determining the mean firing rate, defined as the number of spikes within the bin when the animal's speed exceeded three cm/s, divided by the total dwell times using the same speed threshold. The peak firing rates were identified as the maximal rate on the spatial tuning curve. Spatial selectivity was calculated as one minus the circular variance of the spatial tuning curve. Neurons were classified as place cells when their spatial selectivity exceeded 0.25, and peak firing rates exceeded 1 Hz. To assess the similarity of spatial tuning between pairs of neurons, Spearman correlation coefficients were calculated for the spatial tuning curves. To explore the relationship between levels of synchronization and spatial tuning similarity among pairs of place cells, Spearman correlation coefficients were computed between synchronization strengths of cell pairs and correlation coefficients of their spatial tuning curves.

## Data analysis and statistics

Data analysis was performed using MATLAB (Mathworks, R2019b), and results were presented as standard error of the mean unless otherwise specified. Boxplots were used to depict the median, first quartile, and third quartile, with the middle line, the upper edge, and the lower edge of the box, respectively. The whiskers of the boxplots represented the 5th and 95th percentiles of the distribution.

## Acknowledgements

We express our sincere gratitude to Dr. Luke Lavis for sharing the JF$_{552}$-HaloTag ligand. Special thanks to Mr. Yu-Ting Lin and Ms. Hui-Ching Chen for technical assistance. We are grateful to Drs. Shih-Chieh Lin, Kuo-Hua Huang, Shi-Bing Yang, and Fu-Chin Liu for their insightful comments on the manuscript.

## Additional information

### Funding

| Funder | Grant reference number | Author |
|---|---|---|
| National Science and Technology Council | MOST 110-2320-B-A49A-525 | Bei-Jung Lin |

The funders had no role in study design, data collection and interpretation, or the decision to submit the work for publication.

### Author contributions

En-Li Chen, Data curation; Tsai-Wen Chen, Resources, Software, Validation, Visualization, Methodology; Eric R Schreiter, Resources, Methodology, Writing – review and editing; Bei-Jung Lin, Conceptualization, Resources, Data curation, Software, Formal analysis, Supervision, Funding acquisition, Validation, Investigation, Visualization, Methodology, Writing – original draft, Project administration, Writing – review and editing

### Author ORCIDs

Tsai-Wen Chen ⓘ https://orcid.org/0000-0001-6782-3819
Eric R Schreiter ⓘ https://orcid.org/0000-0002-2864-7469
Bei-Jung Lin ⓘ https://orcid.org/0000-0002-1755-1664

### Ethics

The present study followed institutional guidelines for animal treatment and complied with relevant legislation from the IACUC of National Yang Ming Chiao Tung University. All surgical procedures, behavioral training, and recording rotocols were approved by the IACUC of National Yang Ming Chiao Tung University (IACUC Approval No: 1100427).

Joint Public Review: https://doi.org/10.7554/eLife.96718.6.sa1
Author response https://doi.org/10.7554/eLife.96718.6.sa2

## Additional files

### Supplementary files
MDAR checklist

### Data availability
All data generated or analysed during this study are included in the manuscript and supporting files. Figure 1-source data 1 and Figure 3-source data 1 contain the numerical data used to generate the figures.

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
