## [Editor Report · eLife Assessment]

In this **valuable** study, the authors use a cutting-edge method to perform voltage imaging of CA1 pyramidal cells in head-fixed mice running on a track while local field potentials (LFPs) were recorded in the contralateral hemisphere. The authors provide **solid** evidence of synchronous ensembles of CA1 pyramidal neurons that are associated with contralaterally recorded theta rhythms but not with contralaterally recorded sharp wave-ripples during exploration of a novel environment. The paper will be of interest to scientists who are interested in hippocampal neuronal coding of novel environments, particularly those with experimental questions that can benefit from this cutting-edge imaging technique.

---

## [Referee Report · Joint Public Review]

Summary:

There has been extensive electrophysiological research investigating the relationship between local field potential patterns and individual cell spike patterns in the hippocampus. In this study, the authors used innovative imaging techniques to examine spike synchrony of hippocampal cells during locomotion and immobility states. The authors report that hippocampal place cells exhibit prominent synchronous spikes that co-occur with theta oscillations during exploration of novel environments.

Strengths:

The single cell voltage imaging used in this study is a highly novel method that may allow recordings that were not previously possible using traditional methods.

Weaknesses:

Local field potential recordings were obtained from the contralateral hemisphere for technical reasons, which limits some of the study's claims.

---

## [Author Response]

The following is the authors’ response to the previous reviews

**Joint Public Review:**
Summary:For many years, there has been extensive electrophysiological research investigating the relationship between local field potential patterns and individual cell spike patterns in the hippocampus. In this study, using innovative imaging techniques, they examined spike synchrony of hippocampal cells during locomotion and immobility states. The authors demonstrated that hippocampal place cells exhibit prominent synchronous spikes locked to theta oscillations.Strengths:The single cell voltage imaging used in this study is a highly novel method that may allow recordings that were not previously possible using existing methods.

We thank the reviewer for recognizing the strengths of our study.

Weaknesses:The strength of evidence remains incomplete because of the main claim that synchronous events are not associated with ripples. As was mentioned in previous rounds of review, ripples emerge locally and independently in the two hemispheres. Thus, obtaining ripple recordings from the contralateral hemisphere does not provide solid evidence for this claim. The papers the authors are citing to make the claim that "Additionally, we implanted electrodes in the contralateral CA1 region to monitor theta and ripple oscillations, which are known to co-occur across hemispheres (29-31)" do not support this claim. For example, reference 29 contains the following statement: "These findings suggest that ripples emerge locally and independently in the two hemispheres".

In our previous revisions, we took care to limit our claim to what our data directly supported: that synchronous ensembles of CA1 neurons were not associated with ripple oscillations recorded in the *contralateral* hippocampus. To address reviewer concerns, we changed the Title, modified the Abstract, adjusted relevant text in the Results, and explicitly acknowledged the methodological limitations in the Discussion.

In this round, we further revised the manuscript to directly address the editor’s and reviewer’s remaining concerns:

(1) We replaced the word “surprisingly” with a more neutral “Moreover” to avoid implying that the observed dissociation was unexpected given the use of contralateral recordings.

Introduction (line 67-69):

“Moreover, these synchronous ensembles occurred outside of contralateral ripples (c-ripples) …”

(2) We removed the clause stating that ripples “co-occur across hemispheres”, along with the associated citation to Buzsaki et al. (2003), to avoid potential misinterpretation. The sentence now simply states that we recorded ripple and theta oscillations in the contralateral CA1.

Introduction (line 63-64):

“Additionally, we implanted electrodes in the contralateral CA1 region to monitor theta and ripple oscillations.” (co-occurrence claim removed)

(3) We carefully replaced all mentions of “ripples” in the manuscript with “c-ripples” (i.e., contralateral ripples) to ensure that the scope of our findings is clearly defined and cannot be misinterpreted.

(4) We strengthened the acknowledgment of the methodological limitations in the Discussion.

Discussion (line 528-533):

“While contralateral LFP recordings can capture large-scale hippocampal theta and ripple oscillations, they do not fully reflect ipsilateral-specific dynamics, such as variation in theta phase alignment or locally generated ripple events (Buzsaki et al., 2003; Szabo et al., 2022; Huang et al., 2024). Given that ripple oscillations can emerge locally and independently in each hemisphere, interpretations based on contralateral recordings must be made with caution. Further studies incorporating simultaneous ipsilateral field potential recordings will be essential to more precisely understand local-global network interactions.”

These revisions ensure that our manuscript now presents a consistent and appropriately limited interpretation across all sections. We hope these clarifications address all remaining concerns and accurately reflect the scope of our findings.